# Winners and Losers: *Cordulegaster* Species under the Pressure of Climate Change

**DOI:** 10.3390/insects14040348

**Published:** 2023-03-31

**Authors:** Judit Fekete, Geert De Knijf, Marco Dinis, Judit Padisák, Pál Boda, Edvárd Mizsei, Gábor Várbíró

**Affiliations:** 1Research Group of Limnology, Centre of Natural Science, University of Pannonia, Egyetem St. 10, H-8200 Veszprém, Hungary; 2Centre for Ecological Research, Institute of Aquatic Ecology, Department of Tisza Research, 18/c Bem Sq., H-4026 Debrecen, Hungary; 3Research Institute of Nature and Forest (INBO), Havenlaan 88 bus 73, 1000 Brussels, Belgium; 4CIBIO/InBIO, Centro de Investigação em Biodiversidade e Recursos Genéticos da Universidade do Porto, Instituto de Ciências Agrárias de Vairão, R. Padre Armando Quintas n◦ 7, 4485-661 Vairão, Portugal; 5Departamento de Biologia, Faculdade de Ciências, Universidade do Porto, 4099-002 Porto, Portugal; 6BIOPOLIS Program in Genomics, Biodiversity and Land Planning, CIBIO, Campus de Vairão, 4485-661 Vairão, Portugal; 7ELKH-PE Limnoecology Research Group, Egyetem St. 10, H-8200 Veszprém, Hungary; 8Kiskunság National Park Directorate, Liszt F. St. 19, H-6000 Kecskemét, Hungary; 9Department of Ecology, University of Debrecen, Egyetem Sq. 1, H-4010 Debrecen, Hungary

**Keywords:** species distribution modeling, future distribution, range shifts, Odonata

## Abstract

**Simple Summary:**

Climate change is already affecting biodiversity and will do so even more in the future. As bioclimatic parameters, such as precipitation and temperature, directly or indirectly determine the occurrence of species, the accelerated changes in these variables could have a huge impact on species distributions. In this study, we aimed to use species distribution modeling to predict the potential distribution of the Balkan Goldenring (*Cordulegaster heros*) and the Two-Toothed Goldenring (*C. bidentata*) under recent and future climatic conditions to obtain a more accurate picture of the most suitable areas over time, thus facilitating the planning of conservation projects. According to our results, these montane species are strongly influenced by climatic variables. The models predict that the two species respond differently to changes in bioclimatic variables in the size of the potential range but similarly in range shift.

**Abstract:**

(1) Bioclimatic factors have a proven effect on species distributions in terrestrial, marine, or freshwater ecosystems. Because of anthropogenic effects, the changes in these variables are accelerated; thus, the knowledge of the impact has great importance from a conservation point of view. Two endemic dragonflies, the Balkan Goldenring (*Cordulegaster heros*) and the Two-Toothed Goldenring (*C. bidentata*), confined to the hilly and mountainous regions in Europe, are classified as “Near Threatened” according to the IUCN Red List. (2) Modeling the potential occurrence of both species under present and future climatic conditions provides a more accurate picture of the most suitable areas. The models were used to predict the responses of both species to 6 different climate scenarios for the year 2070. (3) We revealed which climatic and abiotic variables affect them the most and which areas are the most suitable for the species. We calculated how future climatic changes would affect the range of suitable areas for the two species. (4) According to our results, the suitable area for *Cordulegaster bidentata* and *C. heros* are strongly influenced by bioclimatic variables and showed an upward shift toward high elevations. The models predict a loss of suitable area in the case of *C. bidentata* and a large gain in the case of *C. heros*.

## 1. Introduction

Climate change is one of the most serious global environmental threats, which affects the biodiversity of terrestrial, subterranean, marine, and freshwater ecosystems [1]. In freshwater ecosystems, the overall changes in temperature and precipitation lead to the alteration of hydrologic characteristics. Changes in the frequency, duration, and timing of precipitation and snow melt result in the unpredictable alternation of floods and droughts, which have a huge impact on freshwater species and communities [2,3]. Because of the steep topographic slope, the hydrological regime of streams in hilly and mountainous regions largely depends on precipitation regimes, which makes them one of the most vulnerable aquatic ecosystems along with their biota.

Dragonflies are amphibious animals: their larvae develop in standing or running waters, even for several years, while adults are terrestrial with a strong connection to aquatic habitats. They are key species and play an important role in both aquatic and terrestrial communities [4,5,6]. As larvae, they consume other aquatic macro-invertebrates and even vertebrates. As adults, they predate flying insects [4]. Dragonflies and damselflies function as flagship species, both as species worth preserving for their own right and as indicators of a healthy environment (clean water). Many species are sensitive to changes in their habitats and are adapted to various types of freshwaters [4,7]. Biological attributes such as the presence, diversity, and abundance of odonates are used in the bioassessment of freshwater habitats [8,9,10]. Many dragonflies in Europe are assessed as endangered or near-threatened [11]; therefore, acquiring up-to-date knowledge about their occurrence and their population size and trends is very important for their conservation. Populations mainly decline due to habitat destruction or changes in habitat quality through pollution, eutrophication, excessive water abstraction for human consumption and agriculture activities, and droughts.

*Cordulegaster bidentata* Selys, 1843 and *C. heros* Theischinger, 1979 are endemic to Europe, have overlapping ranges, and are known to co-occur sometimes within the same watercourse. Their extent of occurrence encompasses part of the Mediterranean, Alpine, Continental, and Pannonian biogeographical regions [12]. As both species are assessed as “Near threatened” [13,14,15,16], it is of major importance to protect them and conserve their habitats. We hypothesized that it might be worth applying species distribution modeling (SDM) techniques over a European scale dataset to shed light on whether climatic conditions do influence the occurrence of the species.

*C. bidentata* has a wide distribution occurring in Central Europe, extending to Western Europe, where populations are found in France and Belgium, southwards to the mainland of Spain and Italy, and southeastwards to mainland Greece. *C. bidentata* is often confined to springs and small streams in forests [12] reaching its highest abundance in first- and second-order stretches at hilly and mountainous areas (100–2100 m).

Typically, the water flows slowly on calcareous bedrock and is characterized by high hardness and conductivity [17]. Although *C. bidentata* covers a large geographic area, the species is confined to this specific habitat [11] and not considered abundant even in suitable habitats and was assessed as “Near Threatened” on the IUCN Red List based on the narrow habitat preferences and declining population trend [11]. *C. heros* mostly inhabit second-order sections of water courses with low conductivity and hardness [17] but with high dissolved oxygen contents [18]. The distribution of *C. heros* extends from Slovakia, Czechia, and Austria in the north to the Balkan region in the southeast [12], covering a much smaller geographic range than *C. bidentata,* but it is more common within its range and can even be locally abundant. *C. heros* is listed on the Annexes II and IV of the European Habitats Directive and was also assessed as “Near Threatened” on the IUCN Red List, based on a rather small range and declining population trend [11]. The ecological requirements of these two species may overlap, even in the same stream [19,20]. They both occur in shaded hilly and mountainous areas with a balanced water supply; however, the meso- and microhabitat characteristics play a major role in their small-scale patterns. *C. heros* and *C. bidentata* larvae prefer pools with slow water movement and small particle-sized substrates [14,17]. The streams and springs where both species occur are always located in forested areas, resulting in year-round low water temperatures and rather cool ambient temperatures, especially when compared to nearby open habitats. The dissimilarities of larva preferences between the two species could be the results of the different geographical distribution, as *C. heros* is widely distributed in Mediterranean areas, and larvae should have better adaptations to these types of rivers. Hence, Mediterranean streams are often temporal; thus, *C. heros* should cope with at least wider tolerance or niche width for temporal droughts and fluctuations of water and a higher tolerance for low oxygen levels.

Both species have relatively wide distributions, but because of their special habitat preferences, vulnerability, and sensitivity to habitat change, their distribution areas are scattered and endangered by several factors. Due to their long larval development (3–5 years), the most threatening factor is long-term drought, which can lead to local extinction [14,21,22,23]. Microclimate and water regimes are sensitive to large-scale deforestation, which is a major threat to the survival of populations of these species. Depending on the slope, the velocity of the headwaters varies from site to site, and with this also dissolved oxygen, substrate size and deposition [24,25,26], and consequently, the available larval habitat. Therefore, the slope has been generally used as a distributional proxy in studies focused on dragonfly distribution patterns [25,27,28,29,30,31,32]. Despite the relatively similar microhabitat preference, the two species’ geographic distribution significantly differs [12]. This can be the consequence of the macrohabitat and climatic niche preferences of the two species.

This raises the question of how much the occurrence of the two species, *C. heros* and *C. bidentata,* is influenced by bioclimatic and other macrohabitat variables and how the location and size of suitable areas will change under future climate conditions. Furthermore, we investigate how the two closely related species respond to climatic changes. Species can respond to global climate changes by shifting their range to occupy new suitable geographic areas tracking the species’ environmental demands [33]. Many studies have been published about observed or forecasted range shifts in elevational and poleward directions in plant and animal species [34,35].

Species distribution modeling (hereafter SDM) has been widely used in ecological and evolutionary studies across terrestrial, freshwater, and marine realms. These methods can help to identify suitable areas for endangered species, to prioritize conservation actions, to identify areas where invasive species might be present, and also to model the impact of climate change or changes in land use on the distribution of species [36,37,38,39]. There are two main model approaches used for estimating species’ potential occurrence. One is the mechanistic model in which the tolerance of a species to environmental factors is known, and this can be used to calculate where they potentially occur [40,41]. The other is the correlative model approach, in which we do not apply specific tolerance values but rely on a species’ realized distribution to approximate its realized ecological niche [36,42]. Correlative models use data from species observations and selected environmental data with the aim of finding correlations between the known occurrences and the environmental variables and evaluating how the macrohabitat variables determine the distribution of a given species, and testing the result against a portion of training data [25,43,44]. Based on the similarity of environmental variables, the relationship between environment and distribution can be extrapolated across space and time, allowing us to predict a species’ potential distribution at present or in the future and for different geographic regions. Here, we test whether these species with wide niche overlap and distinct degrees of rarity will have similar future distribution patterns in light of climate change.

We decided to use two out of the four available representative concentration pathways (RCP) to simplify calculations. The RCP 4.5 was chosen as it is mentioned as an intermediate or low radiative forcing scenario according to the IPCC 2014 Climate Change 2014: Synthesis Report [45]. It predicts the carbon will peak in the atmosphere around 2040, then it shows a decline. The other scenario we used is a high radiative forcing scenario, the RCP 8.5, which is the worst-case scenario, predicting a continuous rise of CO_2_ in the atmosphere through the 21st century.

The aims of this study are (a) the use of SDMs to describe the current potential distribution of the target species over a European scale dataset; (b) to identify the bioclimatic and topographic macrohabitat drivers of their distributions; and (c) to predict responses to climate change and define the stable, not stable, or potential new distribution area of the species, and the distribution shift. Using niche models, we predict how climate change might affect suitable habitats and impact the distribution of both species. By modeling the potential occurrence of both *Cordulegaster* species, SDMs provide large-scale predictions of present and future suitable areas for the species, which can inform the planning and implementation of monitoring projects and support conservation decision making, particularly in the face of climate change. We hypothesize that changes in the distributional pattern of *C. bidentata* and *C. heros* species will be highly different despite they are closely related species; differences in their habitat requirements can be detected, and these differences will be escalated by climate change.

## 2. Materials and Methods

### 2.1. Occurrence Data

To model *Cordulegaster heros* and *C. bidentata,* occurrence data were collected and digitized from 88 faunistic articles, 6 citizen science surveys, national and international databases (e.g., GBIF, iNaturalist, Odonata.it), and from personal communication, supplemented with our own data from field samplings. The list of data sources can be found in the online Appendix A. The collected dataset covers the whole distribution range of the two species according to the Atlas of the European dragonflies and damselflies [12]. The collected records comprise data on larvae, exuviae, and adults. We used only georeferenced or highly accurate data (inaccuracy less than 5 km, UTM data with a maximum of 10 km resolution). Data that did not fulfill these requirements were omitted from the analyses. The data were digitized using QGIS.

Since occurrence data come from several countries with uneven sampling effort over the species range, our original dataset was spatially resampled to avoid spatial bias [46]. The occurrence data were subsampled, keeping only those records that are at least 10 km apart from each other. The data were selected randomly with 1000 iterations to keep as many records as possible with the least information loss. A total of 108 (of 874) and 743 (of 1839) records for *C. heros* and *C. bidentata,* respectively, were used for building the model.

### 2.2. Environmental Variables

As the main aim of the study is to identify potentially suitable future areas for both *Cordulegaster* species, we used climatic variables available for future scenarios of climate change. The climatic data were supplemented with topographic variables, which are expected to remain unchanged across the relevant time period for our models but make our input dataset more robust. To represent current conditions, eight bioclimatic variables (bio_2: Mean Diurnal Range; bio_4: Temperature Seasonality; bio_10: Mean Temperature of Warmest Quarter; bio_11: Mean Temperature of Coldest Quarter; bio_12: Annual Precipitation, bio_17: Precipitation of Driest Quarter; bio_18: Precipitation of Warmest Quarter; bio_19: Precipitation of Coldest Quarter) and two topographic variables (altitude and slope) were considered. Bioclimatic variables were downloaded from the WorldClim database [47,48] at 5 arcminutes (about 6.5 km at 45° latitude) resolution. Elevation data from SRTM (Shuttle Radar Topography Mission) were downloaded with the ‘getData’ command from the UCDAVIS portal (biogeo.ucdavis.edu), and the slope was calculated with the ‘raster’ package in R [49]. Variables were projected and resampled to the same Coordinate Reference System (CRS WGS84) at a 5 arcminute resolution using the ‘raster’ package in R, and then all of them were cropped and masked to the study area. To define the study area, a buffer was drawn around the species known presence data and calculated with the ‘rgeos’ package [50] gBuffer command with 2.5 decimal degrees. At that size, contiguous areas were formed around the known occurrence data of the species.

We have to bear in mind that, for different species, the ability to disperse and adapt to changes can greatly influence the occurrence patterns of a species. Thus, the predictions calculated by the models can differ greatly from the actual occurrence conditions depending on different dispersal abilities. Two main dispersal types are considered in distribution modeling. One is unlimited dispersal, in which different predictions project suitable habitats over the whole study area, and the other is the no dispersal type, in which the predictions for the future are restricted to the current known range [25,51,52]. In the present study, modeling was performed assuming unlimited dispersal capacity.

When building a model, the selection of predictors has paramount importance as too many variables can increase the risk of collinearity and cause overparameterization, which can distort the accuracy of the models [25]. Therefore, as few variables as possible were selected from the WorldClim database and topographic variables, with the use of correlation analysis to minimize multicollinearity. With the ‘fuzzySim’ package [53] in R, the predictor variables were analyzed with ‘multicol’ to define the multicollinearity and, then filtered with the ‘corSelect’ tool to analyze pairwise correlations between variables specified to Pearson correlation coefficient with 0.8 correlation threshold. The filtered variables were clustered by using a pairwise correlation test, and from the remaining highly correlated groups, the most important variables were selected. A total of six variables per species were kept. The maximum remaining correlation between the selected variables was 0.76 for *C. bidentata* and 0.72 for *C. heros*.

### 2.3. Future Climate Variables

For future conditions, the same bioclimatic variables were considered for the year 2070. Two representative concentration pathways (RCP; 4.5 and 8.5) and three selected Global Circulation Models (GCM) were used for both species: for *C. bidentata,* the BCC-CSM1-1, CCSM4, and the NorESM1-M, and for *C. heros* BCC-CSM1-1, NorESM1-M, and MPI-ESM-LR. Both were derived from the Coupled Model Intercomparison Project Phase 5 (CMIP5; available at WorldClim portal, www.worldclim.org) (accessed on 27 May 2020).

The applied GCMs were selected by using the GCM CompareR tool [54] because circulation models have different accuracies in different areas. For selection, two bioclimatic variables were used, bio_2 and bio_12 for *C. bidentata* and bio_4 and bio_10 for *C. heros*, which proved to be decisive based on preliminary modeling of the species. For the study area, the occurrence area of the two species was marked separately with a rectangle. The tool calculated an ensemble for the available circulation models for RCP 8.5 for the year 2070 in the study area, and then the three GCMs, which showed the least difference from the ensemble could be selected.

### 2.4. Algorithm Selection

The most appropriate model algorithms were selected using the ‘BIOMOD2′ software package version 3.4.6 [55], a package that is an Ensemble Platform for Species Distribution Modeling. In the pre-model selection, 5 widely used model algorithms (GBM: generalized boosting model, ANN: artificial neural network, GLM: generalized linear model, MARS: multiple adaptive regression splines, RF: random forest) were run from the ‘BIOMOD2′ package in five runs and with 5000 randomly selected pseudoabsence data, as absence data were not available for the entire distribution of the species. The data were split into 70% training and 30% testing data sets. The pre-selection process shows that, on average, two machine learning algorithms, the GBM (generalized boosted model) [56,57] in the case of *C. heros* and the RF (random forest [58]) in the case of *C. bidentata*, performed the best according to ROC metric (receiver operating characteristic) [59], for identifying potentially suitable areas for the occurrence of these species; thus, these were used for modeling.

### 2.5. Modeling

The schematic flowchart of the modeling process is shown in Figure 1. Final models were run using GBM and RF algorithms (for *C. heros* and *C. bidentata*, respectively) with default settings in BIOMOD 2. It allows the use of several built-in algorithms for ensemble modeling of species distributions and spatial projection of predictions.

Five pseudoabsence datasets were used in the modeling, and each was composed of 5000 pseudo-absences selected randomly from the study area. A total of 50 models were run per species, consisting of 10 replicates per each of the five pseudoabsence datasets. A threshold-dependent ensemble was performed for species with threshold ROC > 0.8, and the results of the selected models were projected for the present and all future scenarios. The projections for the present conditions show an ensemble of the results of the selected models. The projections were converted to binary maps with maximized sum threshold method (MTSS) in BIOMOD 2 [60]. The binary projections of the three different RCPs were combined with averaging. The projections for the future show the differences in the suitable areas through the scenarios. The loss, gain, and stable areas were calculated with the BIOMOD_RangeSize function. For the projections, the ‘raster’ and ‘maptools’ packages in R [61] were used. Variable importance was averaged across all selected models. The importance of the variables was plotted using the ‘ggplot’ package in R [62]. The future projections were also examined for altitudinal shifts compared to the present projections. The altitude values corresponding to the raster cells defined as suitable for the present and future ensembles were extracted, and their respective averages were compared to quantify the shift.

## 3. Results

### 3.1. Present Distribution

The predicted suitable area for both species under current climatic conditions is presented in Figure 2. The mean predictive accuracy of the selected models based on ROC value was 0.85 in the case of *C. bidentata* (minimum 0.82, maximum 0.87), and 50 models in the ensemble were used. The mean of ROC values in the case of *C. heros* is 0.83 (minimum 0.77, maximum 0.88), and 42 models were used for the ensemble.

### 3.2. Ecological Niche Characteristics of the Species

The importance of environmental variables (permutation-based variable importance values, hereafter: VI) for each model is given in Table 1 and visualized in Figure 3. The most important variables for *C. bidentata* are annual precipitation (bio_12, VI = 0.390) and slope (VI = 0.295). For *C. heros*, temperature seasonality (bio_4, VI = 0.265) and altitude (VI = 0.228) contribute the most among variables.

### 3.3. Future Projections

According to the models, the suitable range of *C. heros* largely increases in both scenarios. The area gain is 37% with RCP 4.5 and 121% with RCP 8.5. while the loss of suitable area is 15.6% and 10.39% in RCP 4.5 and RCP 8.5, respectively (Table 2, Figure 4). In the case of *C. bidentata*, the projections show a large decrease in the contiguous suitable area not only in southern Italy and Sicily but also in northern areas in Germany and Czechia. Altogether the central part of the area remained stable in both scenarios. The loss is 35.1% in RCP 4.5 and 40.8% in the case of RCP 8.5, while the amount of gain in suitable areas is 10.1% and 12.1%, respectively (Table 2, Figure 4).

*C. bidentata* distribution range shows an elevation shift in both future scenarios. The mean altitude is 893 m in the present projections and increases to 1130 m in RCP 4.5 and 1112 m in RCP 8.5. (Figure 4a,c and Figure 5a). The forecasts for suitable areas of *C. heros* also show an upward shift in the elevation values. The mean altitude is 501 m in the present projections, which slightly increases to 544 m in RCP 4.5 and 574 m in RCP 8.5. (Figure 4b,d and Figure 5b).

## 4. Discussion

Climate change is one of the most threatening factors affecting freshwater communities globally [63,64]. Therefore, modeling and understanding the potential impacts of this phenomenon is of paramount importance for the conservation and survival of species [25,65]. Species distribution modeling is a widely used tool to predict the present and future distribution of species, but it also has limitations. Probably the most important in our analyses are the incomplete records of the presence or absence of the species due to the lack of surveys in different areas and due to the low availability of data inherent to large-scale works such as this. To minimize the negative effects of the lack of data coverage, we filtered the training and test data for reliable, spatially declustered, and georeferenced occurrence data, resulting in a considerably sized and accurate dataset (108 and 743 records) and generated pseudoabsence datasets according to them. Another limitation is the lack of high-resolution background data on hydrology and microhabitat at the European scale and for the relevant time period, which we tried to minimize by considering bioclimatic and topographic variables that could act as proxies for these important ecological parameters. Finally, while distribution models are a valuable tool for conservation planning, their predictions are best used when validated with complementary information. While macroclimatic assessments of climate change responses are valuable for the detection of loss of suitable areas, their use for the identification of potential refugia should be complemented with independent data on dispersal corridors and suitable microhabitats.

In this study, distribution models were used to determine which macrohabitat variables best predict habitat suitability for *Cordulagaster heros* and *C. bidentata*. Although our models have low spatial resolution and are best suited for determining macrohabitat preferences, they likely correlate with several ecologically relevant microhabitat factors.

*C. bidentata* prefers spring streams or small upper courses of streams with high oxygen concentrations [17,18,66]. This species is also closely related to limestone and karst springs, whose water supply is irregular and highly correlated with precipitation. As can be seen from our results (Figure 3, Table 1, the most significant topographic variable for *C. bidentata* was slope. A higher slope contributes to an increase in water flow, resulting in higher dissolved oxygen [67]. Among the climatic background variables of this species, an increase in annual precipitation (bio_12) is highly important, as it greatly influences the flow regime of small springs and streams, the key habitat of *C. bidentata*. *C. heros* can be found on hilly and mountainous streams in the Balkans and Central Europe. Based on our models, it can be stated that among the topographic variables, the slope is a less important variable than the altitude for *C. heros*. Among the temperature variables, temperature seasonality plays an important role. The mean temperature of the warmest quarter also has a decisive impact on areas suitable for the species. Among the precipitation variables, the precipitation of the driest quarter proved to be important, which can also be related to the importance of water supply.

The high accuracy of our ensemble models for the present climatic conditions is demonstrated not only by the high ROC but by the fact that the predicted distribution areas highly overlap with the known occurrences, according to our own data, the Atlas of the European dragonflies and damselflies [68] and the IUCN 2021 red list maps [21,69]. Thus, we assume that in the suitable areas where there are no known presence data, the species could theoretically occur given suitable biotic conditions (e.g., predation, competition) and a lack of relevant movement constraints. Our results will definitely contribute to very efficient specific surveys in the predicted areas for these *Cordulegaster* species.

The data for future conditions were projected to predict the effect of climate changes on their distribution and to identify the potentially stable areas where the circumstances are suitable for the species under current and future climatic conditions. Both studied dragonfly species showed high sensitivity to changes in climatic variables. The present area suitable for both species and the predicted area are different as different macrohabitat variables influence their occurrence. The large differences in the future projections can be attributed to the fact that these different bioclimatic variables change at different scales and directions in the future.

Elevational and latitudinal distribution shifts have been observed for many organisms in response to climate change [35]. Several studies focus on the observed and predicted shifts of odonates [29,34,70,71,72]. The forecasts for suitable areas of *C. heros* and *C. bidentata* showed an upward shift in the direction of higher elevation values. *C. heros* can, like many other lotic species with good sipersal ability [68], resist temporary drought periods [14,73]. Therefore, we assume that it will be able to follow the shift of suitable climate. We predict that the main range of *C. bidentata* is likely to remain stable around the Alps, the Pyrenees, and the Dinaric Mountains in the Balkans because in these large and continuous regions the species may move to higher elevations. Where movement to higher elevations is not possible, it is expected that the species will not be able to find suitable areas, such as in northern France, parts of Germany or southern Italy (Figure 4), and will face local extinctions.

*C. heros* belongs to the *boltoni* subgroup within the *Cordulegaster* genus [74] and is closely related to this species [18]. *C. heros* is replaced by *C. boltonii* (Donovan, 1807) in the north and west of its range and by another species of this subgroup, *C. picta* Selys, 1854, in the southeast. The current distribution of *C. heros* is limited by its sister species *C. boltonii* in the rest of Europe as the result of glaciation during the ice ages in Europe and the recolonization. Most likely, *C. heros* or its predecessor survived these cold periods in the southern Balkans and could not move northwards because of the presence of the Carpathian and other mountain ranges, which was still partly covered by snow and ice. Our models forecast the areas suitable for *C. heros* and predict their occurrence in southern Poland and Czechia. As these areas are known to be occupied by *C. boltonii*, the occurrence of *C. heros* is unlikely. The *bidentata* group contains *C. bidentata*, which is widespread in Europe. *C. helladica* Lohmann, 1993 and *C. insignis* Schneider, 1845. *C. bidentata* border the range of *C. helladica* in Greece and *C. insignis* in the eastern Balkans. The species in the groups are parapatric, and this is relevant for species distribution modeling as the edges of the range are not directly a result of environmental constraints but of suitable habitats already occupied by one of its sister species [75]. This often can result in a rather sharp border between the study species and its close relatives. This has an impact on the current distribution but also potential shifts in distribution. For this reason, the predicted distributional areas can be changed, however at an unpredictable way, by interspecific relations. Both model species are threatened by increased drought periods and drying up of habitats, as seen in the southern Balkans, including Romania, or in the south of France, where former populations are known to have gone extinct [21,69]. The extinction could be a result of rainfall deficiency or water scarcity due to extraction, such as in Greece and many parts of the Mediterranean [21,69]. The ongoing long periods of drought due to climate change and their long-term effects, combined with an increase in water abstraction from streams and rivers, increase the extinction risk. Planning and prioritizing conservation management for these species is, therefore, a high priority.

As all European *Cordulegaster* species have narrow habitat preferences, they are often the only odonate species occurring at a specific point. At the more open, sunny patches along the forest streams, the nearly other species occasionally present is the damselfly *Calopteryx virgo,* Linnaeus, 1758, but the larvae have other microhabitat preferences. *Cordulegaster* larvae are burrowers and live in detritus, mud, sand, and between fallen leaves in the water. Larvae of *Calopteryx virgo* prefer small twigs and underwater vegetation where they forage. The habitat of both *Cordulegaster* species is very local in the Balkans, shared with *Caliaeschna microstigma* Schneider, 1845. This species seems to co-habit very well with both *Cordulegaster* species, as in nearly half of the localities, at least one *Cordulegaster* species is present [76].

Our work reveals that the suitable area of these species will shift and/or decrease under future climatic conditions. We assume that other species with similar climatic requirements, such as *Caliaeschna microstigma* in the southern Balkans, would show similar responses in the future. On the one hand, we determined areas that are suitable for these species at present but are threatened by changing climate conditions. On the other hand, we determined the safest areas for survival against the negative effects of climate change, and this way, we could design species protection programs. In these areas, special attention must be paid to reducing the human factors that negatively affect these species’ habitats, such as water extraction, deforestation, and pollution. Intensive monitoring and protection of these safe areas are key to the survival of these species.

## Figures and Tables

**Figure 1 insects-14-00348-f001:**
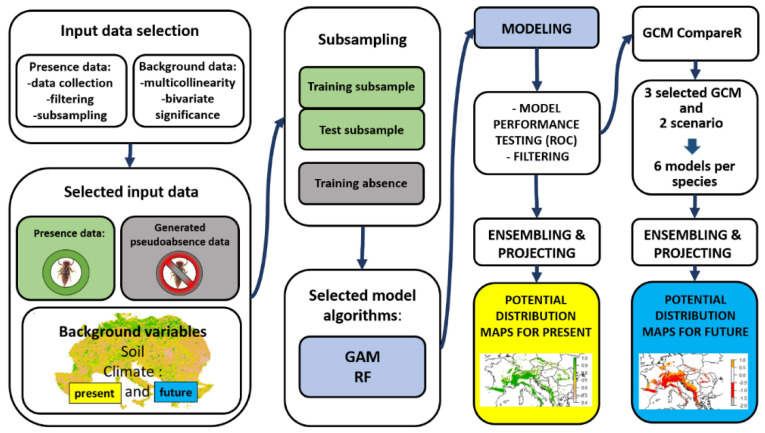
The schematic flowchart of the modeling process.

**Figure 2 insects-14-00348-f002:**
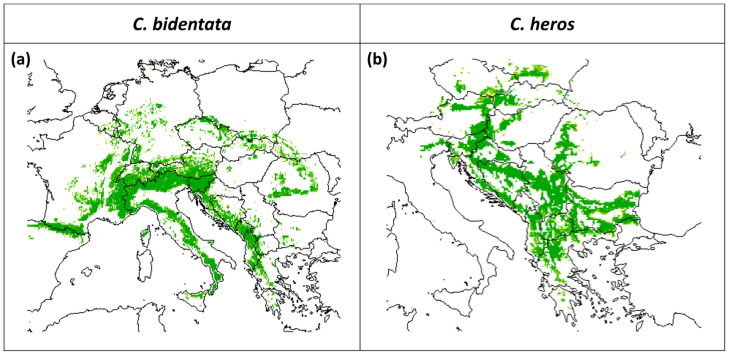
Predicted suitable area under current climatic conditions for (**a**) *C. bidentata* and (**b**) *C. heros*, based on the ensemble of models (scale: predicted potential suitability, green indicates the area suitable for the species).

**Figure 3 insects-14-00348-f003:**
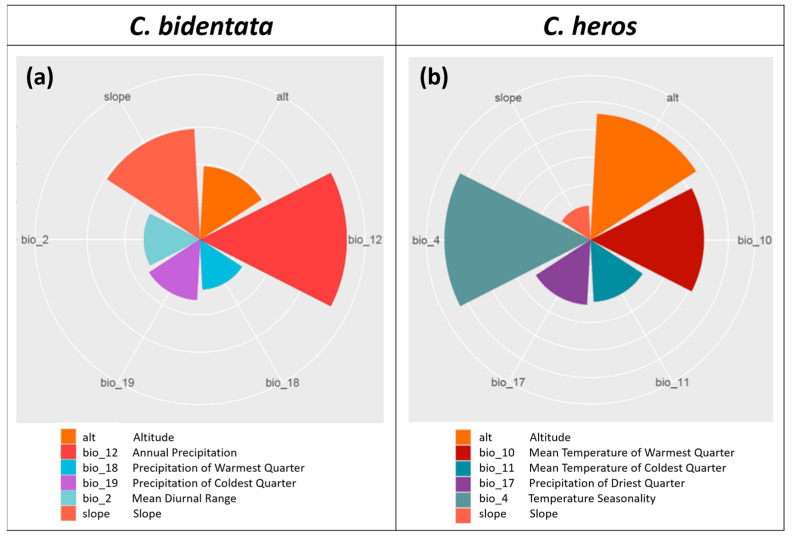
Polar plot of the variable importance scores of (**a**) *C. bidentata*, (**b**) *C. heros.*.

**Figure 4 insects-14-00348-f004:**
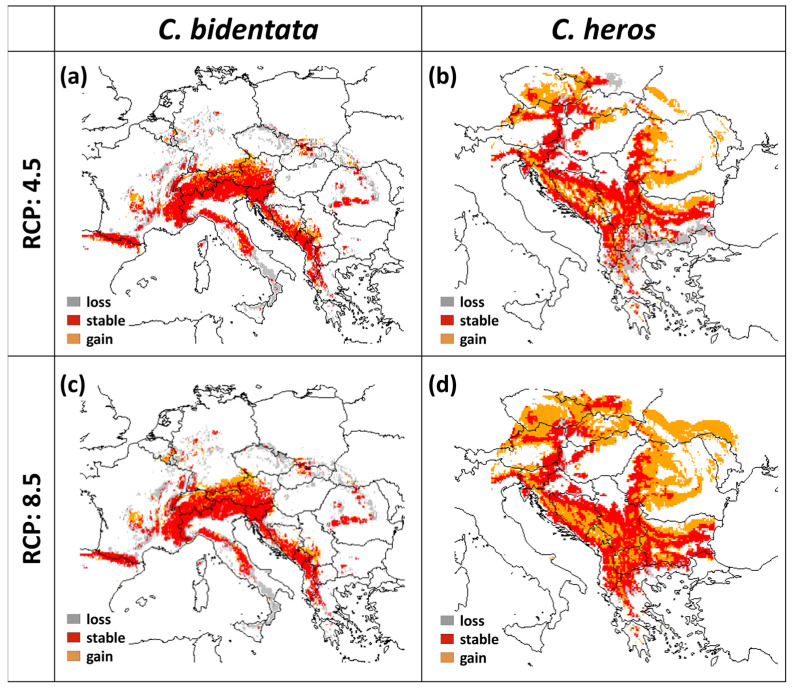
Changes in future projections under different climatic scenarios (for the year 2070) for (**a**) *C. bidentata* under RCP 4.5, (**b**) *C. heros* under RCP 4.5, (**c**) *C. bidentata* under RCP 8.5, and (**d**) *C. heros* under RCP 8.5 (red area: stable, orange area: gain, gray area: loss).

**Figure 5 insects-14-00348-f005:**
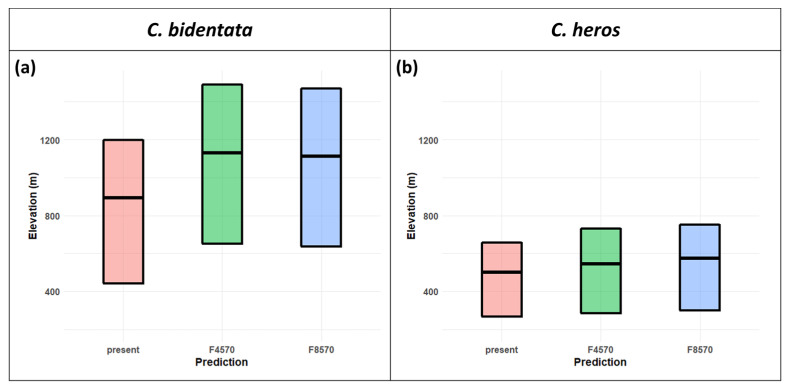
Crossbar visualization of elevation shift for *C. bidentata* (**a**) *C. heros* (**b**) in the future SDM models. The bars show the upper and lower quartile and mean of the values or centroid query from the variables where the projections for the future indicate “stable” or “gain” outcome).

**Table 1 insects-14-00348-t001:** Permutation-based variable importance scores sorted by their importance.

*C. bidentata*		*C. heros*	
Variable	Scores	Variable	Scores
Annual Precipitation (bio_12)	0.390	Temperature Seasonality (bio_4)	0.265
Slope	0.295	Altitude	0.228
Altitude	0.195	Mean Temperature of Warmest Quarter (bio_10)	0.206
Precipitation of Coldest Quarter (bio_19)	0.161	Precipitation of Driest Quarter; (bio_17)	0.118
Mean Diurnal Range (bio_2)	0.149	Mean Temperature of Coldest Quarter (bio_11)	0.113
Precipitation of Warmest Quarter (bio_18)	0.133	Slope	0.062

**Table 2 insects-14-00348-t002:** The changes in the size of suitable area in the number of pixels for the future projections. “Stable” areas were predicted to be suitable in both the present and future, “loss” is when the area is suitable in the present but not in the future, and “gain” is when the area is only suitable in the future.

	*C. bidentata*		*C. heros*	
	RCP45	RCP85	RCP45	RCP85
Loss (Cell Number)	2841	3613	706	392
Stable (Cell Number)	5264	5252	9917	8066
Gain (Cell Number)	816	1073	2375	4969
Percentage of Loss	35.1	40.8	15.6	10.39
Percentage of Gain	10.1	12.1	52.6	131.7
Species Range Change	−25	−28.7	37	121.3

## Data Availability

The script, the model outputs, and the public occurrence data with citations are available online (http://freshwater-ecology.com:3838/CordSDM/) (accessed on 28 March 2023) or on request from the authors.

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
