# Peer review of "Winners and Losers: Cordulegaster Species under the Pressure of Climate Change"

_insects, 2023, doi:10.3390/insects14040348_

Round 1

Reviewer 1 Report

The topic of the study is interesting, but it is necessary to work on the interpretation of several outputs. Especially the presentation of some results and the establishment of clear hypotheses or predictions is desirable. In particular, the descriptions of graphs and tables are very general. The study uses distribution models that are based solely on climate data and lacks other parameters that are more closely related to the quality of microhabitats. However, I understand that such data is not available on a global scale. In the introduction, I would not focus only on the similar requirements of the larvae, but rather mention their differences, because, actually, in the whole study I lack information, why C. heros should spread, while the population of C. bidentata will decrease.

I have inserted other notes directly into the text in the form of notes (sorry about that).

Author Response

Dear Reviewer, 

Thank you so much for your valuable comments, which helped a lot to improve our MS. Please see the attachment for our answers and changes. 

All the best, Judit Fekete on behalf of Co-Authors.

Reviewer 2 Report

Dear authors, 

I want to congratulate for a wonderful and insightful paper about these important odonate species. Overall, I think is an important contribution for freshwater ecosystems conservation. There is a few comments I leave on the manuscript that I think that will make your results more clear and relevant for the readers. 

Author Response

(The authors gave the same response as above.)

Reviewer 3 Report

In their manuscript, Fekete et al. predict the current and future distribution of two Cordulegaster dragonfly species under two distinct climate change scenarios. The manuscript is generally well-written and I found no major problems in either the methodology or the conclusions drawn. In my opinion, the clarity of some sections could be improved and a thorough reading for improving the English would be necessary (please see detailed comments below). The Discussion should be extended to a few important issues: 1) the shift in the lowest elevation of the species, 2) the limitations of the study/methods used, in particular, 2a) the incomplete recording of the species, in terms of presences, 2b) the use of pseudoabsences and 2c) the lack hydrological variables (no stream - no dragonfly, baserock preference etc.), as well as 3) the differences between the two species (why?) and 4) the conservation implications of the shifts. In this last case, the potential impact on the closely related species should also be discussed.

Minor comments (please note that the list of needed English corrections is not exhaustive, I only cherrypicked a few):

Line 35: Please name the two species

Line 39: delete ‘background’

Line 40: should be ‘the most’ and put ‘them’ instead of the second ‘species’

Line 49: ‘the biodiversity’

Line 55: I suggest changing ‘weather fluctuations’ to ‘precipitation regimes’

Lines 54-55: change ‘is largely dependent’ to ‘largely depends’

Lines 62-63: are slightly off-logic, and should be moved to around line 71.

Lines 59-73: The general logic in this section should be straightened to help readers follow the text.

Line 68: ‘of high priority’

Lines 75-84: I am not sure how well this rather anecdotal part fits the article. I am not completely against it but I think it should be improved in style, or even deleted.

Line 91: The distribution of C. bidentata should be described more precisely, similar to that of C. heros.

Line 93: With this elevation range I would not emphasize the importance of altitude in the distribution. 100 m altitude is basically a flatland.

Line 97: delete ‘values of’

Line 102: You may want to merge sentences by replacing ‘C. heros’ with ‘covering’

Line 109 Maybe ‘small-scale’ instead of ‘micro-distribution’

Line 119: delete ‘sensitive’

Line 140: ‘we do not’

Line 149: Consider changing ‘slightly different ecological requirements’ with ‘high niche overlap’

Line 152: ‘to use’ instead of ‘using’ and ‘out of the four’

Line 154: Why did not consider the newer scenarios from the 2021 Assessment Report of IPCC?

Line 156: Choosing an intermediate and the worst-case scenarios may lead to a bias, please have a look at https://www.sciencedirect.com/science/article/pii/S2214629620304655 to see an opinion against going directly to 8.5.

Line 162: ‘SDM’s provide an accurate view of the most suitable areas’ – I would not be that sure, there are heaps of caveats using SDMs. I suggest deleting this.

Line 178: Please, use ‘adults’

Line 179: 5-10km resolution is clearly not low enough to represent microclimate which has been suggested to be important for both species. This needs to be discussed among the limitations.

Line 202: WGS84 is not equidistant and nor does it preserve areas – choosing another CRS may be better.

Table 1. I miss the inclusion of hydrological variables, or at least the presence/absence of streams.

Line 227: What was the VIF threshold?

Line 231: What correlates with what?

Line 245 Delete ‘off’

Line 261: ‘the best’ instead of ‘better’, also close the bracket and delete ’and proved to be the best algorithms’

Line 268-269: Sentence is unclear, please clarify.

Figure 4 – Table 2: One is redundant, keep either the table or the graph (I, personally, prefer the graph). I, however, think it would be better to write out the variables rather than using their codes, or, at least list them in the caption. Thus, the reader does not need to go back and find them in Table 1. This way, on the other hand, may question the need for Table 1.

Fig 6.: This shows a substantial increase in the minimum elevation in the predicted distribution of C. bidentata but less so in the case of C. heros. This should be discussed.

Lines 365-370: These repeat the results, they can be deleted.

Line 366: Yet, it seems that altitude is not highly important for C. heros.

Line 370-371: They absolutely should. I do not think this sentence is informative at all, it may be deleted or, if the authors wanted to point out some important information, rephrased.

Lines 385-387: Logically does not fit here.

Lines 399-404: ‘Where movement to higher elevations is not possible, it is expected that the species will not be able to find suitable areas, like in northern France, parts of Germany or southern Italy (Fig. 5), and will face local extinctions in these areas.’ - sounds nicer

Lines 405-416: The logical flow in this paragraph should be improved, maybe by merging with the next para and reshuffling the sentences.

Line 424: Would C. boltonii not shift?

Line 435: Why this exact species is chosen as an example?

Line 436: Unclear sentence, please rephrase.

Supplementary material: Shiny app does not work properly, disconnects and does not show layers. Judging from my own experience, leaflet and memory usage may be the underlying issue.

Author Response

(The authors gave the same response as above.)
